# Applying the Reasoned Action Approach and Planning to Understand Diabetes Self-Management Behaviors

**DOI:** 10.3390/bs12100375

**Published:** 2022-10-01

**Authors:** Tom St Quinton

**Affiliations:** School of Psychology and Therapeutic Studies, Faculty of Social and Health Sciences, Leeds Trinity University, Leeds LS18 5HD, UK; t.stquinton@leedstrinity.ac.uk

**Keywords:** diabetes management, reasoned action approach, intention, planning, behavior change, blood glucose monitoring

## Abstract

Individuals managing diabetes are required to adhere to self-management behaviors to ensure the optimal regulation of their blood glucose levels. This study examined the psychological determinants underlying three important diabetes self-management behaviors (e.g., physical activity, diet, and blood glucose monitoring) using the reasoned action approach (RAA) and planning. A cross-sectional design was used, with participants (N = 273) completing measures of RAA constructs (e.g., experiential and instrumental attitude, descriptive and injunctive norm, and capacity and autonomy) and planning (e.g., action and control planning) at time 1 and participation in the behaviors one week later at time 2. Regressions showed that RAA constructs accounted for good variance in intention and behavior in all behaviors. Intention towards diet and blood glucose monitoring was significantly predicted by instrumental attitude, injunctive norm, and capacity. Intention towards physical activity was significantly predicted by instrumental attitude, experiential attitude, injunctive norm, capacity, and autonomy. All behaviors were significantly predicted by intention, action planning, and coping planning. Additionally, capacity significantly predicted physical activity and autonomy significantly predicted diet and blood glucose monitoring. Successfully intervening in the influential psychological constructs identified in the study could ensure optimal blood glucose regulation in those managing diabetes.

## 1. Introduction

Diabetes mellitus is a chronic health condition caused by irregular glucose levels and contributes globally to millions of deaths per year [1]. The majority of diabetes cases are classified as Type 2 [2], and predictions are that rates will continue to rise significantly in the future [3]. To manage this condition, those diagnosed are required to make significant lifestyle modifications. Specifically, it is important that blood glucose levels are regulated and maintained through the adoption of several self-management behaviors [4,5]. Regular adherence to such behaviors can reduce diabetes complications [6]. Three self-management behaviors demonstrating improved glycemic control are physical activity [7], healthy diet [8], and regularly monitoring blood glucose levels [9]. In fact, poor glycemic control has been attributed mainly to non-adherence to these behaviors [10].

Poor diabetes management can lead to serious detrimental health consequences, including heart disease, stroke, retinopathy, and early mortality [11,12]. Despite these negative outcomes, a large proportion of those living with diabetes are physically inactive [13,14], do not adhere to a healthy diet [15,16], and struggle to monitor their blood glucose [17,18]. It is therefore important to understand the motivations underlying participation in these self-management behaviors. 

### 1.1. The Reasoned Action Approach

There are many personality, psychosocial, and socio-demographic factors associated with health behavior e.g., [19,20,21]. Theories of social cognition, which focus on the psychological determinants underlying behavior, have been widely used to predict and explain participation in health behavior [22]. Identifying the important and modifiable psychological determinants can facilitate the development of effective health behavior change interventions [23]. A prominent social cognition theory is the reasoned action approach RAA [24]. The RAA is a contemporary extension to the theory of planned behavior [25] and posits intention as the proximal determinant of behavior. Intention is determined by experiential (i.e., feelings and emotions) and instrumental (i.e., cognitive) attitude, injunctive (i.e., the perceived approval of others) and descriptive (i.e., the behavior of others) norms, and capacity (i.e., perceptions of confidence) and autonomy (i.e., perceptions of control).

A meta-analysis conducted by McEachan et al. [26] found the RAA explained 59% and 31% of the variance in intention and behavior, respectively. In relation to diabetes, studies have applied its predecessor, the theory of planned behavior [25], to understand self-management behaviors. Gatt and Sammutt [27] found that attitude, subjective norm, and perceived behavioral control predicted a general measure of self-management behaviors (including foot care, blood glucose monitoring, physical activity, diet, and medication adherence). Specific to physical activity, Ferreira and Pereira [28] found that the model accounted for 45% of the variance in intention and 22% of the variance in behavior. The results showed that attitude and perceived behavioral control were significant predictors of intention, and that intention significantly predicted behavior. Additionally, Boudreau and Godin [29] identified all three antecedents of intention as significant predictors, accounting for 60% of the variance. In relation to diet, White et al. [30] found that theory constructs accounted for 29% of the variance in intentions, with attitude and subjective norm but not perceived behavioral control as significant predictors. Moreover, intention and perceived behavioral control significantly predicted dietary behavior and accounted for 14% of the variance.

Although these studies usefully explained participation in diabetes self-management behaviors, they did not apply the RAA, which, as previously mentioned, is an extension to the theory of planned behavior. Specifically, rather than considering attitude, norm, and control as global constructs, the RAA considers the uniqueness of differentiated components. This enables researchers to establish the specific subcomponents important in the prediction of behavior [26]. The utility of the theory has been demonstrated in studies applying it to explain various health-related behaviors e.g., [23,31,32]. For example, Norman et al. [31] recently adopted the theory to explain participation in different behaviors associated with the spread of COVID-19. However, despite the utility of the theory, no study has adopted the RAA to understand participation in diabetes self-management behaviors. 

### 1.2. The Intention-Behavior Gap

Although the RAA and theory of planned behavior account for impressive variance in intention towards health behavior, research has identified a gap between intention and action [33,34]. That is, despite having positive intentions to engage in a health behavior, behavior is often not undertaken [35]. For example, it has been observed that medium-to-large changes in intention results in only small-to-medium changes in behavior [36]. The journey from intention to behavior often entails overcoming barriers, setbacks, temptations, and unanticipated events [37,38]. Action and coping planning have been suggested to bridge this gap and facilitate intention translation [39]. Action plans are goal-directed plans formed based on what, when, where, and how the behavior will be undertaken. For example, an individual may identify a time and location when blood glucose levels will be monitored during the working day. Coping plans are developed in anticipation of any obstacles that may arise. For example, a person may identify an alternative location to exercise in the event of bad weather. Research has supported the use of planning in promoting health behavior e.g., [40,41,42]. Adopting these self-regulatory strategies could usefully ensure intentions to engage in the three key diabetes self-management behaviors are enacted.

### 1.3. The Present Study

Given the management of diabetes depends heavily on undertaking important health behaviors, it is important that the psychological determinants underlying these behaviors are understood. This can help inform the development of effective interventions promoting the behaviors. The purpose of the study was to therefore explain participation in three key diabetes self-management behaviors using the RAA and self-regulatory determinants. It was predicted that attitude (experiential and instrumental), norms (injunctive and descriptive), and capacity and autonomy would significantly correlate with and predict intention to participate in the behaviors. It was also expected that the behaviors would be significantly correlated with and predicted by intention, capacity, autonomy, and action and coping planning. 

## 2. Materials and Methods

### 2.1. Participants

A sample of 273 participants were recruited. To be included in the study, participants had to be over the age of 18 years, diagnosed as a Type 2 diabetic, and monitor their blood glucose using a finger prick test.

### 2.2. Design and Procedure

A prospective correlational design was used with assessments taken at baseline and one week later. Participants were recruited from Diabetes UK support groups in the North of England, UK. The support groups were provided with recruitment materials and were asked to circulate the information to their members. Links to the study, which was conducted online using Online Surveys, were included in the recruitment materials. Once the survey had been accessed, participants read detailed information about the study and their involvement in it. Participants willing to participate then provided informed consent and completed the first questionnaire. Following completion, participants were contacted again one week later to complete behavioral measures. A pseudo code was generated to match data across time-points. Ethical approval was granted from the Faculty Research Ethics Committee before data collection commenced. 

### 2.3. Measures

Demographics: participants reported their age in years, gender, and ethnicity.

Psychological determinants: the psychological determinants associated with physical activity, healthy eating, and blood glucose monitoring were assessed. RAA items were developed following the guidelines provided by Fishbein and Ajzen [24]. Participants completed two items measuring experiential attitude (e.g., To what extent would doing each of the behaviors listed below over the next week be unpleasant/pleasant, r = : 0.89, *p* < 0.01 for diet; 0.92, *p* < 0.01 for physical activity; 0.83, *p* < 0.01 for blood glucose monitoring), two items measuring instrumental attitude (e.g., To what extent would doing each of the behaviors listed below over the next week be valuable/worthless, r = : 0.84, *p* < 0.001 for diet; 0.73, *p* < 0.01 for physical activity; 0.86, *p* < 0.01 for blood glucose monitoring), one item measuring descriptive norms (e.g., I think that most people who are important to me would perform the behaviors listed below over the next week, definitely no/definitely yes), one item measuring injunctive norms (e.g., To what extent would other people disapprove or approve of you doing each of the behaviors listed below over the next week? disapprove/approve), two items measuring capacity (e.g., How confident are you that you could do each of the behaviors listed below over the next week? not at all confident/very confident, r = : 0.92, *p* < 0.001 for diet; 0.86, *p* < 0.001 for physical activity; 0.72, *p* < 0.01 for blood glucose monitoring), two items measuring autonomy (e.g., I have complete control over undertaking each of the behaviors listed below over the next week, strongly disagree/strongly agree, r = : 0.77, *p* < 0.01 for diet; 0.72, *p* < 0.05 for physical activity; 0.75, *p* < 0.05 for blood glucose monitoring), and two items measuring intention (e.g., I am likely to perform each of the behaviors below over the next week, strongly disagree/strongly agree, r = : 0.93, *p* < 0.001 for diet; 0.94, *p* < 0.001 for physical activity; 0.89, *p* < 0.001 for blood glucose monitoring). In addition to RAA constructs, planning was measured using similar items recently adopted by Hamilton et al. [43]. Specifically, four items measured action planning (e.g., In the next week, I have made a plan when to perform physical activity, strongly disagree/strongly agree, Cronbach α = : 0.74 for diet; 0.83 for physical activity; 0.76 for blood glucose monitoring) and four items measured coping planning (e.g., I have made a plan what to do if something interferes with my goal of being physically active, strongly disagree/strongly agree, Cronbach α = : 0.72 for diet; 0.88 for physical activity; 0.79 for blood glucose monitoring). All items used 7-point scales varying in direction.

Self-management behaviors: compliance with the self-management behaviors was reported one week later. The Diabetes Self-Care Activities Measure [44] was used to assess participation in the three behaviors. The questionnaire includes 11 items assessing engagement in five behaviors over the previous 7 days. Items assessing foot care and smoking were not included meaning eight items were used in the study. Specifically, four items measured diet (e.g., How many of the last seven days have you followed a healthful eating plan? Cronbach’s α = 0.89), two items measured physical activity (e.g., On how many of the last seven days did you participate in at least 30 min of physical activity? r = 0.91, *p* < 0.001), and two items measured blood glucose monitoring (e.g., On how many of the last seven days did you test your blood sugar? r = 0.93, *p* < 0.001). 

### 2.4. Analysis

Data analyses were undertaken using IBM SPSS (version 27.0). Descriptive statistics were generated on demographic variables, psychological constructs, and the self-management behaviors. Bivariate correlations were conducted between RAA constructs, planning, and the behaviors. Following this, a multiple linear regression was conducted in relation to intention to engage in the behaviors. Experiential attitude, instrumental attitude, injunctive norm, descriptive norm, autonomy, and capacity were the independent variables and intention was the dependent variable. These were conducted separately for each self-management behavior. A hierarchical regression was then conducted in relation to behavior. Intention, autonomy, and capacity were entered at Step 1, and action planning and coping planning were entered at Step 2. These were conducted separately for each self-management behavior. With power = 0.80, alpha = 0.05, and five and six predictor variables in the hierarchical and linear regressions, respectively, the sample size obtained was sufficient to detect small to medium effect sizes.

## 3. Results

### 3.1. Descriptive Statistics and Correlations

The sample consisted of 273 participants (Male: *n* = 147; Age: *M* = 57.31 years, *SD* = 9.42, Range = 20–78; White: *n* = 267, Black: *n* = 3, Asian: *n* = 3). Of those completing baseline measures, 256 participants completed follow-up assessments. There were no significant differences between completers and non-completers related to demographics and baseline psychological constructs.

The means, standard deviations, and correlations between RAA constructs, planning, and behaviors can be seen in Table 1. As was expected, there were significant positive correlations between RAA constructs and intention in relation to all behaviors. Only experiential attitude did not significantly correlate with intention to engage in blood glucose monitoring. Therefore, stronger instrumental and experiential attitude, descriptive and injunctive norms, and capacity and autonomy was associated with higher intentions to engage in physical activity and diet, and stronger instrumental attitude, descriptive and injunctive norms, and capacity and autonomy was associated with higher intentions to engage in blood glucose monitoring. Moreover, and as was predicted, intention, capacity, autonomy, action planning, and coping planning was significantly positively correlated with all three behaviors. Therefore, stronger capacity, autonomy, action planning, and coping planning was associated with higher participation in physical activity, diet, and blood glucose monitoring.

### 3.2. Regression Analyses Predicting Physical Activity

Multiple regression analyses indicated instrumental attitude, experiential attitude, injunctive norm, descriptive norm, capacity, and autonomy explained 52% of the variance in intention, *F*(6, 266) = 50.76, *p* < 0.001. All constructs except descriptive norm were significant predictors (see Table 2). Hierarchical regression analyses indicated intention, capacity, and autonomy explained 48% of the variance in behavior, *F*(3, 252) = 80.10, *p* < 0.001. Intention and capacity were significant predictors (Step 1, Table 2). The inclusion of action planning and coping planning significantly increased the amount of explained variance in behavior (Δ*R*^2^ = 0.07; *F*(2, 250) = 20.15, *p* < 0.001), with all constructs explaining 55% of the variance in behavior, *F*(5, 250) = 63.43, *p* < 0.001 (Step 2, Table 2). All constructs except autonomy were significantly associated with behavior, with intention demonstrating the largest influence on behavior.

### 3.3. Regression Analyses Predicting Healthy Diet

Multiple regression analyses indicated instrumental attitude, experiential attitude, injunctive norm, descriptive norm, capacity, and autonomy explained 51% of the variance in intention, *F*(6, 266) = 48.75, *p* < 0.001. Instrumental attitude, injunctive norm, and capacity were significant predictors (see Table 3). Hierarchical regression analyses indicated intention, capacity, and autonomy explained 49% of the variance in behavior, *F*(3, 252) = 83.61, *p* < 0.001. Intention and autonomy were significant predictors (Step 1, Table 3). The inclusion of action planning and coping planning significantly increased the amount of explained variance in behavior (Δ*R*^2^ = 0.04; *F*(2, 250) = 11.35, *p* < 0.001), with all constructs explaining 53% of the variance in behavior, *F*(5, 250) = 58.83, *p* < 0.001 (Step 2, Table 3). All constructs except capacity were significantly associated with behavior, with intention demonstrating the largest influence on behavior.

### 3.4. Regression Analyses Predicting Blood Glucose Monitoring

Multiple regression analyses indicated instrumental attitude, experiential attitude, injunctive norm, descriptive norm, capacity, and autonomy explained 60% of the variance in intention, *F*(6, 266) = 69.60, *p* < 0.001. Instrumental attitude, injunctive norm, and capacity were significant predictors (see Table 4). Hierarchical regression analyses indicated intention, capacity, and autonomy explained 38% of the variance in behavior, *F*(3, 252) = 51.94, *p* < 0.001. Intention was the only significant predictor (Step 1, Table 4). The inclusion of action planning and coping planning significantly increased the amount of explained variance in behavior (Δ*R*^2^ = 0.06; *F*(2, 250) = 12.57, *p* < 0.001), with all constructs explaining 43% of the variance in behavior, *F*(5, 250) = 39.05, *p* < 0.001 (Step 2, Table 4). All constructs except capacity were significantly associated with behavior, with intention demonstrating the largest influence on behavior.

## 4. Discussion

The management of diabetes depends heavily on individual engagement in health behaviors including physical activity, diet, and blood glucose monitoring. However, research has found inconsistent uptake of these behaviors. The study adopted the RAA and self-regulatory constructs to understand participation in these behaviors. Findings showed the model accounted for significant variance in intention and behavior in all three behaviors. The determinants underlying intention and behavior were similar for diet and glucose monitoring. The inclusion of action and coping plans significantly contributed to the prediction of all three behaviors.

### 4.1. Explaining Intention to Participate in the Self-Management Behaviors

In line with predictions, the findings demonstrated instrumental attitude, injunctive norm, and capacity significantly predicted participation in all three self-management behaviors. This suggests knowledge about the behaviors, the perceived approval of important referents, and confidence to undertake the behaviors are important motivators underlying physical activity, diet, and glucose monitoring. Knowledge of diabetes and the role these behaviors play in glucose homeostasis are central to managing the condition [45,46]. However, people may hold misconceptions about the importance of these behaviors in managing diabetes [47]. The importance of perceived social approval has been demonstrated previously e.g., [48,49,50]. For example, Pereira et al. [49] identified the significant role of family members in providing diabetes social support and Siopis et al. [50] identified friends and health professionals as important influences. With regards to capacity, the determinant showed the largest effect on intention to undertake two of the behaviors (diet and glucose monitoring). Studies have previously demonstrated the importance of efficacy beliefs and perceived ability in managing diabetes e.g., [51,52,53]. For example, Adu et al. [51] found participants were only moderately able to interpret blood glucose patterns and Booth et al. [52] identified issues with consuming correct food types and portion sizes. In addition to these determinants, results showed experiential attitude and autonomy also predicted intention to engage in physical activity, with the former demonstrating the largest effect of all predictors. Research relating to physical activity more generally has found that the feelings, emotions, and pleasures associated with the behavior are important influences [54,55]. Specific to diabetes, Blicher-Hansen et al. [56] found that those managing diabetes identified elevated moods and the joys associated with physical activity as important factors. Autonomy suggests the importance of having control over physical activity. Previous work has found those managing diabetes cite factors such as fatigue [57], a lack of time [58], and poor weather [51] as inhibiting control to be physically active.

Despite the predictions, not all determinants significantly contributed to the prediction of each of the three behaviors. However, this is in accordance with the RAA in that the relative influence of determinants can vary depending on the behavior and population investigated [24]. It is interesting that descriptive norm, despite significantly correlating with intention to engage in all three behaviors, did not significantly predict any behavioral intentions. An explanation for this could be the behaviors under investigation; McEachan et al. [26] found descriptive norm was more strongly correlated with health-risk than health protecting behaviors.

### 4.2. Explaining Participation in the Self-Management Behaviors

As was expected, participation in the three behaviors was significantly predicted by intention, with the construct demonstrating the largest effect on all behaviors. This suggests that intention is an important determinant of physical activity, diet, and glucose monitoring in those managing diabetes. This has been supported in previous work examining these behaviors e.g., [27,28,30]. Behavior was also predicted by autonomy (in diet and glucose monitoring) and capacity (in physical activity). Ensuring that intentions are supplemented with requisite levels of control and perceived capability is therefore important.

Despite the importance of these RAA constructs, as was expected, the inclusion of action and coping plans contributed significantly to the prediction of all self-management behaviors. Planning has been shown to effectively promote health behavior [41,42,59], including the behaviors investigated here e.g., [40,60,61]. Planning helps foster intentions by resolving some of the problems encountered when initiating and persisting with intentions. Specifically, action plans facilitate in starting an intention and coping plans foresee any obstacles or challenges that may be encountered during the performance of behavior [62,63]. Adopting these self-regulatory strategies can help bridge the gap between intention and behavior [41].

## 5. Future Perspectives and Limitations

The findings have important implications for interventions designed to promote adherence to diabetes self-management behaviors. All behaviors would benefit from education emphasizing their advantages, importance, and why they should be adopted [64,65]. It is also important that those with diabetes receive the social support and approval of significant others to engage in the behaviors [66]. This is especially important given that other people may not be aware of the importance of engaging in health behaviors for the diabetic person [67]. Finally, it is important that diabetics perceive that they possess the capacity to undertake these behaviors. In addition to these determinants, physical activity interventions should also target experiential attitude and autonomy.

To modify these determinants, different strategies and behavior change techniques can be used. There are many available and surveying the current literature can facilitate this decision. For example, a meta-analysis conducted by Cradock et al. [68] found that ‘instruction on how to perform a behavior’, ‘behavioral practice/rehearsal’, and ‘demonstration of the behavior’ demonstrated the most effectiveness in interventions targeting physical activity and diet in diabetes patients. Van Rhoon et al. [69] identified ‘social support (unspecified)’, ‘goal setting (outcome/behavior)’, ‘feedback on behavior’, and ‘self-monitoring of outcome(s) of behavior’ as effective techniques used in technological interventions targeting similar behaviors.

The importance of intention suggests targeting these underlying determinants is an important step in promoting these behaviors. However, this is only the first step, and interventions should also encourage the use of planning. Specifically, interventions should utilize action planning by promoting the specification of ‘when’, ‘where’, and ‘how’ each self-management behavior will be undertaken. Moreover, individuals should adopt coping plans by planning in anticipation of any obstacles that could prevent them from enacting these behaviors. Developing plans to enact these behaviors could usefully translate positive intentions into self-management behaviors.

The study was not without limitations. First, the adopted cross-sectional design cannot establish causation of psychological determinants. Second, recruiting from a diabetes support group may have introduced selection bias. Third, the sample explicitly comprised Type 2 diabetics meaning results may not generalize to those managing Type 1 diabetes. However, given that the majority of those diagnosed with diabetes are classified as Type 2, the study findings are highly relevant. Finally, the study relied on participants self-reporting their participation in the behaviors. Studies employing objective behavioral measures would be beneficial. Notwithstanding these limitations, the study predicted a number of important diabetes self-management behaviors using a contemporary behavior change theory. This enabled a good understanding of the behaviors, and the study findings provide important avenues for future interventions.

## 6. Conclusions

The study applied the RAA to explain participation in three key diabetes self-management behaviors. Interventions should focus on modifying the important psychological determinants underlying intention to engage in each behavior. Successfully modifying these determinants could engender more positive intentions towards the behaviors. In addition to focusing on RAA constructs, planning appears to also be an important self-regulatory strategy fostering intention enactment. Therefore, intention to participating in these behaviors should be supplemented with planning strategies. Interventions intervening on these constructs could bring about a positive change in these three key diabetes self-management behaviors. This, in turn, would ensure the optimal regulation of blood glucose levels in those living with diabetes.

## Figures and Tables

**Table 1 behavsci-12-00375-t001:** Bivariate correlations between RAA constructs, planning, and physical activity, diet, and glucose monitoring (N = 273).

	M	*SD*	1	2	3	4	5	6	7	8	9	10
1. Instrumental attitude	5.11 5.21 6.11	1.89 1.83 1.37		0.41 ** 0.63 ** −0.04	0.26 ** 0.37 ** 0.77 **	0.03 0.40 ** 0.30 **	0.34 ** 0.56 ** 0.65 **	0.57 ** 0.54 ** 0.29 **	0.45 ** 0.60 ** 0.67 **	0.34 ** 0.32 ** 0.27 **	0.35 ** 0.29 ** 0.28 **	0.33 ** 0.42 ** 0.39 **
2. Experiential attitude	4.23 4.78 1.84	1.97 1.94 1.57			0.33 ** 0.34 ** −0.05	0.20 ** 0.34 ** 0.16 **	0.66 ** 0.67 ** 0.01	0.32 ** 0.63 ** 0.11	0.62 ** 0.53 ** 0.02	0.46 ** 0.37 ** 0.0	0.40 ** 0.38 ** 0.03	0.47 ** 0.36 ** 0.0
3. Injunctive norm	4.29 5.19 6.22	2.06 2.07 1.23				0.20 ** 0.55 ** 0.30 **	0.35 ** 0.35 ** 0.70 **	0.33 ** 0.32 ** 0.37 **	0.43 ** 0.49 ** 0.71 **	0.38 ** 0.27 ** 0.27 **	0.30 ** 0.25 ** 0.28 **	0.38 ** 0.30 ** 0.40 **
4. Descriptive norm	3.66 4.06 3.37	2.19 2.17 2.15					0.35 ** 0.38 ** 0.25 **	0.18 ** 0.29 ** 0.25 **	0.27 ** 0.40 ** 0.22 **	0.26 ** 0.31 ** −0.11	0.21 ** 0.30 ** −0.03	0.26 ** 0.25 ** 0.05
5. Capacity	3.45 4.30 5.85	2.09 1.93 1.48						0.40 ** 0.74 ** 0.50 **	0.60 ** 0.61 ** 0.72 **	0.47 ** 0.42 ** 0.28 **	0.39 ** 0.42 ** 0.32 **	0.55 ** 0.54 ** 0.51 **
6. Autonomy	4.44 4.33 5.37	2.23 1.94 1.77							0.46 ** 0.52 ** 0.37 **	0.31 ** 0.40 ** 0.10	0.29 ** 0.42 ** 0.09	0.34 ** 0.56 ** 0.32 **
7. Intention	3.79 5.17 6.32	2.08 1.82 1.15								0.70 ** 0.59 ** 0.44 **	0.62 ** 0.51 ** 0.38 **	0.68 ** 0.66 ** 0.60 **
8. Action planning	3.45 3.95 5.74	2.03 2.19 1.64									0.81 ** 0.49 ** 0.64 **	0.66 ** 0.55 ** 0.46 **
9. Coping planning	3.31 4.08 5.50	1.93 2.13 1.79										0.62 ** 0.49 ** 0.43 **
10. Behavior	3.68 5.00 6.33	2.04 2.01 1.27										

Note. ** *p* < 0.01; the top row represents physical activity; the middle row represents diet; the bottom row represents blood glucose monitoring.

**Table 2 behavsci-12-00375-t002:** Regression analyses predicting physical activity intention (N = 273) and behavior (N = 256).

Independent Variable	B	SE B	*β*	Δ*R*^2^	*R* ^2^
Intention					
Instrumental attitude	0.15	0.06	0.14 *		0.52 ***
Experiential attitude	0.33	0.06	0.31 ***		
Injunctive norm	0.15	0.05	0.15 **		
Descriptive norm	0.08	0.04	0.09		
Capacity	0.22	0.06	0.22 ***		
Autonomy	0.12	0.05	0.13 *		
Behavior					
Step 1				0.48 ***	0.48 ***
Intention	0.53	0.06	0.54 ***		
Capacity	0.22	0.06	0.23 ***		
Autonomy	0.00	0.05	0.00		
Step 2				0.07 ***	0.55 ***
Intention	0.27	0.07	0.27 ***		
Capacity	0.20	0.05	0.20 ***		
Autonomy	0.00	0.04	0.01		
Action planning	0.24	0.08	0.24 **		
Coping planning	0.18	0.08	0.17 *		

Note. * *p* < 0.05, ** *p* < 0.01, *** *p* < 0.001.

**Table 3 behavsci-12-00375-t003:** Regression analyses predicting diet intention (N = 273) and behavior (N = 256).

Independent Variable	B	SE B	*β*	Δ*R*^2^	*R* ^2^
Intention					
Instrumental attitude	0.30	0.06	0.31 ***		0.51 ***
Experiential attitude	0.02	0.06	0.02		
Injunctive norm	0.21	0.05	0.24 ***		
Descriptive norm	0.01	0.05	0.01		
Capacity	0.30	0.07	0.32 ***		
Autonomy	0.02	0.06	0.03		
Behavior					
Step 1				0.49 ***	0.49 ***
Intention	0.56	0.06	0.49 ***		
Capacity	0.04	0.07	0.04		
Autonomy	0.29	0.07	0.28 ***		
Step 2				0.04 ***	0.53 ***
Intention	0.41	0.07	0.36 ***		
Capacity	0.03	0.07	0.03		
Autonomy	0.24	0.07	0.23 ***		
Action planning	0.17	0.05	0.19 ***		
Coping planning	0.12	0.05	0.12 *		

Note. * *p* < 0.05, *** *p* < 0.001.

**Table 4 behavsci-12-00375-t004:** Regression analyses predicting blood glucose monitoring intention (N = 273) and behavior (N = 256).

Independent Variable	B	SE B	*β*	Δ*R*^2^	*R* ^2^
Intention					
Instrumental attitude	0.17	0.05	0.20 **		0.60 ***
Experiential attitude	0.03	0.03	0.05		
Injunctive norm	0.27	0.06	0.29 ***		
Descriptive norm	−0.02	0.02	−0.03		
Capacity	0.30	0.05	0.38 ***		
Autonomy	0.01	0.03	0.02		
Behavior					
Step 1				0.38 ***	0.38 ***
Intention	0.53	0.08	0.48 ***		
Capacity	0.11	0.07	0.13		
Autonomy	0.07	0.04	0.10		
Step 2				0.06 ***	0.43 ***
Intention	0.40	0.08	0.36 ***		
Capacity	0.10	0.07	0.11		
Autonomy	0.08	0.04	0.11 *		
Action planning	0.13	0.05	0.16 *		
Coping planning	0.10	0.05	0.14 *		

Note. * *p* < 0.05, ** *p* < 0.01, *** *p* < 0.001.

## Data Availability

The data that support the findings of this study are available from the corresponding author upon reasonable request.

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
