# Peer review of "Applying the Reasoned Action Approach and Planning to Understand Diabetes Self-Management Behaviors"

_behavsci, 2022, doi:10.3390/bs12100375_

Round 1
Reviewer 1 Report
The manuscript examined the psychological determinants underlying three important diabetes self-management behaviors using the reasoned action approach and planning. I think this manuscript can be published in the journal of Behavioral Sciences.
Author Response
The manuscript examined the psychological determinants underlying three important diabetes self-management behaviors using the reasoned action approach and planning. I think this manuscript can be published in the journal of Behavioral Sciences.
Response: thank you for taking the time to review the manuscript and for the recommendation.
Reviewer 2 Report
Dear Authors,
I believe that this paper reflects a good job done. However, I have some contributions for improvement and learning for the continuation of this study.
The theoretical review presented is quite succinct.
The research is finished. I believe it is okay. So, it is not necessary to go deeper. Nevertheless, It is suggested to always keep that in mind in order to avoid bias.
To facilitate understanding, it is suggested to think about the possibility of inserting a table with the variables (name and concept) and their respective indicators, as well as a model that visually represents these relationships.
Regarding "Design and Procedure", it is suggested to justify the choice of the sample.
Another contribution would be to present a figure that represents the methodology.
I also would like to give you a suggestion for the next one: in terms of methodology, it would be better to have between 3 and 6 indicators.
The authors could use Partial Least Square (PLS). It is a statistical technique that evaluates the influences simultaneously.
The first reference (final list) seems to be out of the format.
Finally, it is suggested to improve "conclusions". The authors can present the theoretical and practical implications, limitations, and suggestions for future research.
Author Response
I believe that this paper reflects a good job done. However, I have some contributions for improvement and learning for the continuation of this study.
The theoretical review presented is quite succinct.
The research is finished. I believe it is okay. So, it is not necessary to go deeper. Nevertheless, It is suggested to always keep that in mind in order to avoid bias.
Response: Thank you for these comments. Please see below either some changes made to the manuscript or considerations to be taken onboard for the future.
To facilitate understanding, it is suggested to think about the possibility of inserting a table with the variables (name and concept) and their respective indicators, as well as a model that visually represents these relationships.
Response: Thank you for this suggestion. This is something that will be adopted going forward.
Regarding "Design and Procedure", it is suggested to justify the choice of the sample.
Response: Please see the following inclusion has been included on page 5 representing statistical power: “With power =.80, alpha = .05, and five and six predictor variables in the hierarchal and linear regressions, respectively, the sample size obtained was sufficient to detect small to medium effect sizes.”
Another contribution would be to present a figure that represents the methodology.
Response: Thank you for this suggestion, although I am unsure exactly what this means. If this is a requirement then I would be happy to include in a further revision.
I also would like to give you a suggestion for the next one: in terms of methodology, it would be better to have between 3 and 6 indicators.
Response: Thank you for this suggestion, which will be noted for next time.
The authors could use Partial Least Square (PLS). It is a statistical technique that evaluates the influences simultaneously.
Response: This was something that was considered and could be used in the future.
The first reference (final list) seems to be out of the format.
Response: Thank you for pointing this out. Please see this is now amended.
Finally, it is suggested to improve "conclusions". The authors can present the theoretical and practical implications, limitations, and suggestions for future research.
Response: Thank you for this comment. Please note these are outlined in the ‘Future Perspectives and Limitations’ section, and some are also mentioned in the conclusion. However, the implications have been strengthened in the conclusion. Specifically, it has been explicitly stated that interventions should first focus on RAA constructs to modify intention. Following this, it is suggested that interventions target planning to facilitate intention translation and promote behavioral change. Hopefully these amendments are satisfactory.
Reviewer 3 Report
The authors in the study title “applying the reasoned action approach and planning to understand diabetes self-management behaviors) examined three diabetes self-management behaviors (physical activity, diet, and blood glucose monitoring) using the reasoned action approach (RAA) and planning. I recommend publishing the manuscript. However, main issues have to be addressed.
1- What are the inclusion criteria used to select the 273 participants rather than they diagnosed as type 2 diabetes?
2- Regarding the age of participants, authors mentioned they are above 18 years old, what is the maximum age?
3- Did authors consider the gender difference between the participants, if yes, please write this in the text?
4- The authors should mention if the participants had medical conditions rather than the diabetes type 2, which may affect on the results.
5- How authors determine the average time to complete the experiment? Did authors carry out Pilot test?
6- Authors need to discuss diabetes self-management factors that mentioned in the literatures and clarify if the diabetes self-management behaviours are discussed before or not
Authors mentioned one of the limitations is “Finally, the study relied on participants self-reporting 141 their participation in the behaviors”. Authors can mention how can improve this point in the future
Author Response
The authors in the study title “applying the reasoned action approach and planning to understand diabetes self-management behaviors) examined three diabetes self-management behaviors (physical activity, diet, and blood glucose monitoring) using the reasoned action approach (RAA) and planning. I recommend publishing the manuscript. However, main issues have to be addressed.
Response: Thank you for the suggestions and the recommendation. Please see responses to suggestions below.
1- What are the inclusion criteria used to select the 273 participants rather than they diagnosed as type 2 diabetes?
Response: Please note the following on page 3 specifies the inclusion criteria: “To be included in the study, participants had to be over the age of 18 years, diagnosed as a Type 2 diabetic, and monitor their blood glucose using a finger prick test.”
2- Regarding the age of participants, authors mentioned they are above 18 years old, what is the maximum age?
Response: Please see the range has been included on page 5 (20-78).
3- Did authors consider the gender difference between the participants, if yes, please write this in the text?
Response: Thank you for this comment. The study did not consider gender differences and instead focused on motivators across genders. These differences could certainly be an interesting study moving forward.
4- The authors should mention if the participants had medical conditions rather than the diabetes type 2, which may affect on the results.
Response: Thank you for this comment. Measures of other conditions were not taken for the study. However, on reflection understanding comorbidities would have been useful to know. This will be considered going forward.
5- How authors determine the average time to complete the experiment? Did authors carry out Pilot test?
Response: There was no pilot test as the study only required the completion of two questionnaires. The researcher had conducted similar research previously, and so had experience of the time requirements.
6- Authors need to discuss diabetes self-management factors that mentioned in the literatures and clarify if the diabetes self-management behaviours are discussed before or not
Response: Thank you for this comment. Please see the importance of these three behaviors is stated in the opening two paragraphs on pages 1-2. Specifically, the literature suggests these are the behaviors important to regulate glucose levels but are nevertheless infrequently performed. The focus of the study was therefore on these behaviors
.
Authors mentioned one of the limitations is “Finally, the study relied on participants self-reporting 141 their participation in the behaviors”. Authors can mention how can improve this point in the future
Response: Please see the following inclusion on page 13 recommending the use of objective measures: “Studies employing objective behavioral measures would be beneficial.”
Reviewer 4 Report
I congratulate the author for his interesting work. However, I would like to ask for minor improvements before printing:
1. It would be useful to write down whether the author calculated the sample size. There were 273 respondents in the study, and quite a few variables were included in the regression analysis. I think the potential reader needs to understand the statistical power of the analysis.
2. The author does a good job of revealing the relevance of the study in the Introduction, but I think the theoretical review would be better if information about the personality, psychosocial and socio-demographic factors associated with healthy intentions and behaviors were added. The following papers are relevant, and the author may want to take a look (the author does not have to cite all of these works, but it would be good to mention certain factors):
Bigot, A., Banse, E., Cordonnier, A., & Luminet, O. (2021). Sociodemographic, cognitive, and emotional determinants of two health behaviors during SARS-CoV-2 outbreak: An online study among French-speaking Belgian responders during the spring lockdown. Psychologica Belgica, 61(1), 63-78. https://doi.org/10.5334/pb.712
Fathnezhad-Kazemi, A., Aslani, A., & Hajian, S. (2021). Association between perceived social support and health-promoting lifestyle in pregnant women: A cross-sectional study. Journal of Caring Sciences, 10(2), 96-102. https://doi.org/10.34172/jcs.2021.018
Monds, L., Maccann, C., Mullan, B., Wong, C., Todd, J., & Roberts, R.D. (2015). Can personality close the intention-behavior gap for healthy eating? An examination with the HEXACO personality traits. Psychology Health and Medicine, 21(7), 1-11. https://doi.org/10.1080/13548506.2015.1112416
Otsuka, T., Konta, T., Sho, R., Osaki, T., Souri, M., Suzuki, N., Kayama, T., & Ueno, Y. (2021). Factors associated with health intentions and behaviour among health checkup participants in Japan. Scientific Reports, 11, 19761. https://doi.org/10.1038/s41598-021-99303-y
Zolotareva, A., Shchebetenko, S., Belousova, S., Danilova, I., Tseilikman, V., Lapshin, M., Sarapultseva, L., Makhniova, S., Sarapultseva, M., Komelkova, M., Hu, D., Luo, S., Lisovskaya, E., & Sarapultsev, A. (2022). Big Five traits as predictors of a healthy lifestyle during the COVID-19 pandemic: Results of a Russian cross-sectional study. International Journal of Environmental Research and Public Health, 19, 10716. https://doi.org/10.3390/ijerph191710716
3. The author has described the limitations of the study in Implications. I think it would be better if the author highlighted the "Strengths and Limitations" or "Limitations and Future Perspectives" section.
Author Response
I congratulate the author for his interesting work. However, I would like to ask for minor improvements before printing:
- It would be useful to write down whether the author calculated the sample size. There were 273 respondents in the study, and quite a few variables were included in the regression analysis. I think the potential reader needs to understand the statistical power of the analysis.
Response: Thank you for this comment. Please see mentioning of sample size has been included on page X: With power =.80, alpha = .05, and five and six predictor variables in the hierarchal and linear regressions, respectively, the sample size obtained was sufficient to detect small to medium effect sizes.
- The author does a good job of revealing the relevance of the study in the Introduction, but I think the theoretical review would be better if information about the personality, psychosocial and socio-demographic factors associated with healthy intentions and behaviors were added. The following papers are relevant, and the author may want to take a look (the author does not have to cite all of these works, but it would be good to mention certain factors):
Bigot, A., Banse, E., Cordonnier, A., & Luminet, O. (2021). Sociodemographic, cognitive, and emotional determinants of two health behaviors during SARS-CoV-2 outbreak: An online study among French-speaking Belgian responders during the spring lockdown. Psychologica Belgica, 61(1), 63-78. https://doi.org/10.5334/pb.712
Fathnezhad-Kazemi, A., Aslani, A., & Hajian, S. (2021). Association between perceived social support and health-promoting lifestyle in pregnant women: A cross-sectional study. Journal of Caring Sciences, 10(2), 96-102. https://doi.org/10.34172/jcs.2021.018
Monds, L., Maccann, C., Mullan, B., Wong, C., Todd, J., & Roberts, R.D. (2015). Can personality close the intention-behavior gap for healthy eating? An examination with the HEXACO personality traits. Psychology Health and Medicine, 21(7), 1-11. https://doi.org/10.1080/13548506.2015.1112416
Otsuka, T., Konta, T., Sho, R., Osaki, T., Souri, M., Suzuki, N., Kayama, T., & Ueno, Y. (2021). Factors associated with health intentions and behaviour among health checkup participants in Japan. Scientific Reports, 11, 19761. https://doi.org/10.1038/s41598-021-99303-y
Zolotareva, A., Shchebetenko, S., Belousova, S., Danilova, I., Tseilikman, V., Lapshin, M., Sarapultseva, L., Makhniova, S., Sarapultseva, M., Komelkova, M., Hu, D., Luo, S., Lisovskaya, E., & Sarapultsev, A. (2022). Big Five traits as predictors of a healthy lifestyle during the COVID-19 pandemic: Results of a Russian cross-sectional study. International Journal of Environmental Research and Public Health, 19, 10716. https://doi.org/10.3390/ijerph191710716
Response: Thank you for this suggestion. To account for these additional influences, a sentence has been included in the paragraph introducing theories of social cognition. However, given these models primarily consider psychological influences, the focus is on such processes. Please see the following inclusion on page X: “There are many personality, psychosocial and socio-demographic factors associated with health behavior (e.g., Bigot et al., 2021; Monds et al., 2016; Otsuka et al., 2021). Theories of social cognition, which focus on the psychological determinants underlying behavior, have been widely used to predict and explain participation in health behavior (Conner & Norman, 2015)”.
- The author has described the limitations of the study in Implications. I think it would be better if the author highlighted the "Strengths and Limitations" or "Limitations and Future Perspectives" section.
Response: As per the suggestion, this has been amended to ‘Future Perspectives and Limitations’.